# Pre-Administration of PLX-R18 Cells Protects Mice from Radiation-Induced Hematopoietic Failure and Lethality

**DOI:** 10.3390/genes13101756

**Published:** 2022-09-28

**Authors:** Vidya P. Kumar, Shukla Biswas, Gregory P. Holmes-Hampton, Michal Sheleg, Sasha Stone, Betre Legesse, Racheli Ofir, Sanchita P. Ghosh

**Affiliations:** 1Armed Forces Radiobiology Research Institute, Uniformed Services University of the Health Sciences, Bethesda, MD 20889, USA; 2Pluristem Ltd., Haifa 3508409, Israel

**Keywords:** acute radiation syndrome, hematopoietic radiation injury, placenta-derived stromal cells, PLX-R18, prophylactic countermeasure

## Abstract

Acute Radiation Syndrome (ARS) is a syndrome involving damage to multiple organs caused by exposure to a high dose of ionizing radiation over a short period of time; even low doses of radiation damage the radiosensitive hematopoietic system and causes H-ARS. PLacenta eXpanded (PLX)-R18 is a 3D-expanded placenta-derived stromal cell product designated for the treatment of hematological disorders. These cells have been shown in vitro to secrete hematopoietic proteins, to stimulate colony formation, and to induce bone marrow migration. Previous studies in mice showed that PLX-R18 cells responded to radiation-induced hematopoietic failure by transiently secreting hematopoiesis related proteins to enhance reconstitution of the hematopoietic system. We assessed the potential effect of prophylactic PLX-R18 treatment on H-ARS. PLX-R18 cells were administered intramuscularly to C57BL/6 mice, −1 and 3 days after (LD70/30) total body irradiation. PLX R18 treatment significantly increased survival after irradiation (*p* < 0.0005). In addition, peripheral blood and bone marrow (BM) cellularity were monitored at several time points up to 30 days. PLX-R18 treatment significantly increased the number of colony-forming hematopoietic progenitors in the femoral BM and significantly raised peripheral blood cellularity. PLX-R18 administration attenuated biomarkers of bone marrow aplasia (EPO, FLT3L), sepsis (SAA), and systemic inflammation (sP-selectin and E-selectin) and attenuated radiation-induced inflammatory cytokines/chemokines and growth factors, including G-CSF, MIP-1a, MIP-1b, IL-2, IL-6 and MCP-1, In addition, PLX-R18 also ameliorated radiation-induced upregulation of pAKT. Taken together, prophylactic PLX-R18 administration may serve as a protection measure, mitigating bone marrow failure symptoms and systemic inflammation in the H-ARS model.

## 1. Introduction

People required to enter radiation-contaminated areas, such as first responders, military personnel, or clean-up crews, are at risk of developing severe adverse health effects associated with hematopoietic acute radiation syndrome (H-ARS) [1]. Depending on the dose of radiation, symptoms can occur in the period of hours to weeks, for the H-ARS subsyndrome, hypoplasia or aplasia of the bone marrow is expected which can lead to a decreased ability to fight infection, bleeding, pour wound healing, and ultimately death [1]. Exposure to ionizing radiation increases the long-term risk for development of cancer and cataracts, as well as infertility and birth defects [2,3]. To ensure the safety of the war fighter or first responder, it is critical to develop radioprotective countermeasures that can be administered prior to entry into the contaminated site to minimize the harmful effects of radiation exposure and reduce the risk of developing ARS; however, no pre-exposure radioprotective countermeasure has yet been approved by the United States Food and Drug Administration (FDA).

The radiation countermeasures approved by the FDA [1] can successfully treat H-ARS after symptoms develop, but are unsuitable for pre-exposure administration. The four approved countermeasures for H-ARS are Neupogen^®^ (granulocyte colony-stimulating factor (G-CSF), filgrastim) [4], Neulasta^®^ (PEGylated G-CSF, pegfilgrastim) [5], Leukine^®^ (granulocyte macrophage colony-stimulating factor (GM-CSF), sargramostim) [6], and Nplate^®^ (thrombopoietin (TPO) analog, romiplostim) [7]; Neupogen, Neulasta, and Leukine stimulate white blood cells and Nplate stimulates platelets. However, no prophylactic countermeasure has been approved by FDA to date.

Cell therapies have been investigated as both pre-exposure mitigators and post-exposure therapeutics for H-ARS [8,9,10]. The advantage of cell therapies for prophylactic use is that cells adapt their activity based on their environment, and can thus demonstrate a therapeutic effect in the case of radiation injury while having minimal effect on a non-injured person. Furthermore, they can act as a slow-release system, secreting therapeutic molecules over time as necessary.

Various cell therapies have been evaluated for ARS treatment [11,12]. Mesenchymal stromal cells (MSCs) in particular show promise for therapeutic treatment of H-ARS [12]. Administration of MSCs to mice after radiation exposure has been shown to mitigate effects of ARS [13,14,15,16], including increased survival in animals treated with these cell therapies, accelerated recovery of complete blood cell counts, and progenitor bone marrow cells. In some cases this recovery showed improvement over the FDA approved therapy of G-CSF (Neupogen) [14]. Despite the utility demonstrated by these studies for the potential of cell therapies in treating ARS, they are not without drawbacks from a logistical standpoint. Often cell therapies must be administered intravenously which in a mass casualty situation would cause a massive hindrance in the triage of patients, as well as additional supplies constraints in a situation where resources could be scarce.

PLacenta eXpanded-R18 (PLX-R18), a 3D-expanded placenta-derived stromal cells are MSC-like cells that induce hematopoietic system recovery after radiation in animal studies. In rhesus macaques, administration of PLX-R18 following LD50 total body irradiation (TBI) increased 45-day survival by up to 72% and reduced the duration of critical neutropenia, thrombocytopenia, and anemia by up to 1.5 days, compared to vehicle-treated animals [17]. In mice, PLX-R18 treatment after TBI has been shown to significantly increase survival, red blood cell counts, hemoglobin levels, and platelet counts by post-TBI day 23, and to lead to increased white blood cell counts and improved BM colony formation capabilities [18,19,20,21,22]. The potential of PLX-R18 to stimulate hematopoietic recovery is supported by the first-in-human Phase I trial of PLX-R18 (NCT03002519), in which patients with incomplete hematopoietic recovery following hematopoietic cell transplantation (HCT) who were treated with PLX-R18 showed significant improvement in hemoglobin, platelets, and neutrophils compared to baseline [22,23]. In vitro, PLX-R18 cells have been shown to induce bone marrow cell migration and colony formation [21]. It is hypothesized that the therapeutic effect of PLX-R18 is primarily due to the cells’ secretion of a variety of cytokines in response to environmental stimuli such as those arising in a situation of bone marrow failure [21]. Cytokines secreted by PLX-R18 include interleukin (IL)-6, IL-8, G-CSF, and monocyte chemoattractant protein-1 (MCP-1), which are known to induce hematopoietic proliferation, migration, and differentiation [24,25,26,27].

PLX-R18 cells have been demonstrated to have a favorable safety profile and to be well-tolerated in humans [22]. These results are supported by positive toxicology and biodistribution results in rhesus macaques [17] and in a GLP study in mice (unpublished): no significant adverse PLX-R18-related effects were observed, and PLX-R18 cells were localized to the injection site and nearly or entirely cleared from the system by study end (6–14 weeks).

PLX-R18 cells are developed by Pluristem Ltd. (Haifa, Israel), and they are derived predominantly from the fetal portions of full-term placentae donated following Cesarean section and are expanded ex vivo. PLX-R18 cells are characterized by their high expression of MSC markers [28], but exhibit decreased capability to differentiate in vitro into osteocytes and adipocytes compared to BM-derived MSCs, and have limited population doubling capacity [21,29]. As PLX-R18 cells lack expression of HLA class II molecules (HLA-DP, DQ, and DR) or co-stimulatory markers (CD80, CD86, and CD40), they can be used as an allogeneic therapy without HLA matching. PLX-R18 cells are produced in a cGMP facility at industrial scale, cryopreserved and shipped on demand, and are administered by a simple intramuscular (IM) injection; thus making it ideal for ARS treatment scenarios. Thus far, PLX-R18 has only been studied in its capacity as a radiomitigator administered after radiation exposure [21]. In this study, we will investigate for the first time the prophylactic use of PLX-R18 administered prior to radiation exposure to protect the hematological system from radiation damage.

## 2. Materials and Methods

### 2.1. PLX-R18 Preparation and Administration

PLX-R18 cells were produced and supplied by Pluristem Ltd. (Haifa, Israel). In PLX-R18 production, the cells are digested from donated placental tissue, initially expanded in large tissue culture flasks, then transferred to bioreactors for further controlled 3D-expansion on non-woven fiber-made carriers, and finally harvested and cryopreserved. Immediately prior to administration, the cells are thawed, and administered intramuscularly (IM) at a dose of 2 million cells/mouse in 0.1 mL PlasmaLyte (20 million cells/mL), distributed as 0.05 mL/thigh. Mice received two such doses, administered 4 days apart, as described later.

### 2.2. Mice

Male C57BL/6J mice (10–12 weeks old) were purchased from Jackson Laboratory (Bar Harbor, ME, USA) and housed in an air-conditioned facility at the Division of Laboratory Animal Resources (DLAR) of the Uniformed Services University of Health Sciences (USUHS), which is accredited by the Association for Assessment and Accreditation of Laboratory Animal Care International (AAALAC). The mice were housed in rooms with a 12/12 h light/dark schedule, and were fed with Harlan Teklad rodent diet and acidified water ad libitum. All animal procedures were performed according to protocols approved by the USUHS Institutional Animal Care and Use Committee (IACUC) [30,31,32].

### 2.3. Safety Study Design

To determine the safety of PLX-R18 in C57BL/6J mice, a 14-day (after the last administration) acute toxicity study was undertaken in 13-week old mice [33]. Three groups of C57BL/6J male mice (n = 5/group) were used for the study: Naïve, PlasmaLyte, and PLX-R18. In the PLX-R18 group, 2 million cells/mouse were administered IM as described above on Day 1 and again on Day 5; in the PlasmaLyte group, PlasmaLyte (the PLX-R18 vehicle) was administered instead of PLX-R18 using an identical administration regimen. Naïve group was used as an untreated control. The animals were monitored for signs of acute toxicity 1–4 h after each PLX-R18/PlasmaLyte administration, and for signs of chronic toxicity until 14 days after the second dose (Day 19 since the start of the study). On Days 1 (4 h after the first dose), 5 (4 h after the second dose), 7, 9, 14, and 18, weights of all animals were recorded and blood was collected from animals by submandibular incision for complete blood count (CBC) analysis. At the end of the study on Day 19, animals were euthanized, blood was collected for serum chemistry, and gross necropsy was conducted on animals from all three groups.

### 2.4. Total Body Irradiation (TBI)

Mice were irradiated bilaterally in AFRRI’s Cobalt-60 γ irradiation facility in well-ventilated plexiglass radiation boxes (8 animals per box) at an approximate dose rate of 0.6 Gy/min (exact dose rate specific to each study is mentioned in the respective sections). The alanine electron spin resonance (ESR) dosimetry system was used to measure dose rates of water in cores of mouse phantoms. The radiation field was uniform within ±2%. The animals were returned to their cages with food and water within a 60 min period [31,32].

### 2.5. Survival Study in TBI Model Using Cobalt-60 γ Radiation

Fourteen-week-old C57BL/6 male mice were exposed to 8 Gy (0.56 Gy/min dose rate) Cobalt-60 TBI. Prior to study initiation, mice were weighed, animals outside ±10% of the mean weight were excluded, and animals were randomly distributed as 24 animals per group. The animals were administered either PLX-R18 (2 million cells/dose) or the vehicle PlasmaLyte, twice IM as described above: one day prior to TBI (Day −1) and 3 days post-TBI (Day 3). After irradiation, animals were monitored thrice daily for 30 days, which is the critical period during which animals experience radiation injury symptoms and peak mortality. Surviving animals were euthanized on Day 30. This study was repeated for confirmation and cumulative survival data was plotted as Kaplan–Meier plots and statistical significance of the survival difference was determined by log-rank test using GraphPad Prism7 software.

### 2.6. Hematopoietic Recovery Study in TBI Model Using Cobalt-60 γ Radiation

C57BL/6 male mice (14 weeks old) were distributed into 4 groups. Groups 1 and 3 received PlasmaLyte and groups 2 and 4 received PLX-R18 (2 million cells/dose/mouse) on days-1 and +3 with respect to TBI. Groups 1 and 2 received 0 Gy (and were treated the same way as described in the TBI section but with no exposure to radiation) and groups 3 and 4 were subjected to 8 Gy TBI at 0.63 Gy/min. At different time points various tissues and blood was collected for various analyses (CBC, as described below in the subsections.

#### 2.6.1. CBC and Differential Analyses

Roughly 20 µL of blood was collected by puncturing the submandibular vein in Minivette POCT (20 µL, EDTA containing from Sarstedt Inc., Newton, NC, USA). CBCs and differential analyses were performed using a HESKA Element HT5 Analyzer system (Cuattro Veterinary, Loveland, CO, USA) [33].

#### 2.6.2. Blood Collection and Tissue Collection

After making sure that the animal was completely anesthetized using isoflurane (Abbott Laboratories, Chicago, IL, USA) (no response on pinching), the animal was moved to an individual nose cone for continued anesthesia. Blood was collected by cardiac puncture and transferred to serum separator tubes and was allowed to clot for at least 30 min at room temperature and then centrifuged at 1500× *g* for 10 min. The serum was then aliquoted into micro-centrifuge tubes and stored at −80 °C until further use in ELISA and Luminex assay. Following blood collection, the animal was euthanized by cervical dislocation and sterna (in 10% Formalin), spleen (snap frozen in liquid nitrogen and stored at −80 °C until further use) and femurs (in PBS) were collected. Sterna were analyzed for histopathology, femurs were processed as mentioned below for colony-forming unit (CFU) assay and spleen was processed for RT-PCR and Western blotting as described below.

#### 2.6.3. Sternal Histopathology

The sterna collected were fixed in 10% neutral buffered formalin in 1:20 volume (tissue: fixative) for at least 24 h and up to a maximum of 7 days. These fixed sterna were then decalcified in 12–18% EDTA (neutral) for 3 h, after which the specimens were dehydrated and embedded in paraffin. Longitudinal 5 μm sections were cut and stained with Carazzi’s Hematoxylin and Eosin (H&E). The bone marrow was qualitatively evaluated for cellularity within the sternebrae and megakaryocytes were counted by averaging the number of cells per 10 (40×) high-powered fields (HPFs).

#### 2.6.4. Hematopoietic Progenitor Clonogenic Assay (CFU Assay)

Femurs were collected and their bone marrow isolated by flushing with 1.5 mL Iscove’s MDM +2% FBS for mouse cells (Stem Cell Technologies Inc., Vancouver, BC, Canada) per femur. Cells from femurs from three animals were pooled together, followed by washing twice with Iscove’s MDM + 2% FBS. The cell count was adjusted to 2 × 10^5^ cells/mL and added to Mouse MethoCult^®^ (Stem Cell Technologies, Vancouver, BC, Canada). Each sample was plated in duplicate in 35 mm cell culture dishes using a syringe and blunt end needles. Plated cells were incubated for 14 days in a humidified incubator at 37 °C and 5% CO_2_. Colonies were counted 14 days after plating. Fifty or more cells were considered one colony [33].

#### 2.6.5. ELISA to Quantify Radiation Injury and Endothelial Biomarkers from Serum

Mouse Erythropoietin (EPO), Serum Amyloid (SAA), Flt3 ligand (Flt3L), E-Selectin, sP-Selectin and matrix metallopeptidase 9 (MMP-9) (R&D Systems, Minneapolis, MN, USA) and Mouse Procalcitonin (PCT) ELISA kit (My Biosource, San Diego, CA, USA) were used to quantify from serum, following vendor protocols. The data was represented as mean ± SEM for n = 4 mice in duplicate.

#### 2.6.6. Luminex Assay to Study Cytokines

Bio-Plex Pro Mouse Cytokine 23-plex assay kit (BioRad, Hercules, CA, USA) was used to analyze serum preserved at −80 °C as per the vendor’s protocol [34] Fifty microliters of magnetic beads provided in the kit were used with 50 μL of standards or diluted (1:4) serum samples. The magnetic plate washer, Bio-Plex Pro was used for recommended number of washes. Quantity of the biotinylated detection antibody cocktail and streptavidin-phycoerythrin was as per vendor’s protocol. The plate was then immediately read on Luminex 200 (BioRad, Hercules, CA, USA) and the Bio-Plex Manager 6.1.1 was used for data acquisition.

#### 2.6.7. Total RNA Extraction from Spleen Tissue and Autoimmune Pathway Analysis

Harvested spleens were immediately snap frozen in liquid nitrogen and stored at −80 °C. The frozen tissue was homogenized by brief sonication in ice and total RNA was extracted using mirVana total RNA isolation kit (Life Technologies, Frederick, MD, USA; #AM 1560) following the manufacturer’s protocol. RNA yield and quality were analyzed on a NanoDrop spectrophotometer ND-1000 (ThermoFisher Scientific Inc., Rockville, MD, USA). cDNA was synthesized from the RNA using RT^2^ First Strand kit (Qiagen, Germantown, MD, USA). The Autoimmune Pathway (PAMM-077ZC) RT PCR array was performed using a QuantStudio 3 PCR Machine, and the analysis was completed using the Qiagen GeneGlobe Data Analysis Center online [35].

#### 2.6.8. Western Blot Analysis from Spleen Tissue

Frozen spleen tissue was ground with mortar and pestle on dry ice, resuspended in cold RIPA buffer (Sigma-Aldrich, St Louis, MO, USA) containing a protease inhibitor and phosphatase inhibitor cocktail and then homogenized. The protein supernatant was collected and the protein concentration was determined using a bicinchoninic acid protein assay kit (BCA assay, Pierce, Rockford, IL, USA). The proteins were diluted in NuPAGE LDS Sample Buffer (4×) and denatured at 95 °C for 10 min. Following this, 50 µg of protein (in total 30 µL volume) was loaded per lane in a 4–12% Bis-Tris 18-well Criterion XT precast polyacrylamide gel (BioRad, Hercules, CA, USA). The gel was run at 70 V on the Criterion Midi format Cell™ (BioRad), then transferred using Trans-blot Turbo transfer system onto PVDF membrane. Blocking and incubation of membranes with primary and secondary antibodies were carried out using the Starting Block (ThermoFisher). Blots were washed using 0.05% TBST and TBS. The following antibodies were used: β-Actin (8457), JNK (9252), p-JNK (4668), AKT (9272), and pAKT (9271) from Cell Signaling. β-actin was used as the protein loading control. Densitometric analyses of protein bands were performed using Image Lab 5.2.1 Software from BioRad [33].

## 3. Results

### 3.1. PLX-R18 Administered Twice Four Days Apart Found to Be Safe

PLX-R18 was administered to mice in two 2 million-cells/dose, four days apart (on Days 1 and 5) in the safety study (Appendix A). Control groups received PlasmaLyte instead of PLX-R18 (PlasmaLyte group), or no injections (Naïve group). Differences in peripheral complete blood counts (CBC) were primarily limited to differences on Days 1–5 of the study (Appendix A), as follows. Though animals in the PLX-R18 and PlasmaLyte groups showed lower white blood cell, platelet, and lymphocyte counts than the Naïve group on Day 1, by Day 5 the cell counts were similar between the three groups. Additionally, the PLX-R18 group showed significantly higher neutrophil counts on Day 1 than the other two groups, but had similar counts to the other groups by Day 7. No significant changes were observed in other parameters between the groups. No signs of acute toxicity (e.g., decreased activity, squinting eyes, hunching, labored breathing, injection site swelling), chronic toxicity (e.g., significant body weight loss (Appendix A), decreased activity, hunched posture, labored breathing), mortality, or any other abnormal clinical signs of toxicity were observed throughout the study. Gross necropsy at the end of the study did not show any difference between the groups. No differences between the groups were observed in the renal and hepatic serum chemistry panel (Alkaline phosphatase (ALKP), Aspartate transaminase (AST), Alanine aminotransferase (ALT), Creatinine, total protein and blood urea nitrogen (BUN)) assessed from blood collected at the end of the study (Appendix A). In summary, data from this acute toxicity study suggested that PLX-R18 can be safely administered to mice at two 2 million cells/dose, 2 doses four days apart to evaluate its radioprotective efficacy.

### 3.2. PLX-R18 Protects Mice from Lethal Dose of Irradiation

In the 30-day survival study, administration of PLX-R18 one day prior to and three days following lethal irradiation (8 Gy) dramatically increased 30-day survival compared to administration of the vehicle PlasmaLyte (log-rank test *p* < 0.0001; Figure 1). PLX-R18-treated animals showed 81% survival, whereas PlasmaLyte treated animals showed 23% survival. The survival efficacy was confirmed in duplicate independent experiments (n = 24/group/study). This data demonstrated that PLX-R18 was effective in protecting mice from lethal dose of radiation when administered starting the 1st dose at 24 h prior to radiation with a 2nd dose at day 3 post-radiation exposure.

### 3.3. PLX-R18 Administration Accelerated Hematological Recovery in Irradiated Mice

In all groups subjected to 8 Gy TBI, white blood cell, platelet, neutrophil, and lymphocyte counts declined as expected, usually reaching a nadir on Day 9, after which point recovery began. Administration of PLX-R18 significantly increased white blood cell (*p* = 0.0047), platelet (*p* = 0.0070), neutrophil (*p* = 0.0003), and lymphocyte (*p* = 0.0025) counts on Day 21 post-irradiation, compared to administration of PlasmaLyte vehicle (Figure 2). Likewise, PLX-R18 administration significantly increased neutrophil counts on Day 9 (*p* = 0.017) and platelet (*p* = 0.016) and lymphocyte (*p* = 0.046) counts on Day 14, compared to administration of PlasmaLyte vehicle. In the non-irradiated groups, no significant differences in blood cell counts were observed over time or between the PLX-R18 and PlasmaLyte groups, showing no adverse effect of the PLX-R18 administration. The CBC profile data indicated that PLX-R18 pre-treatment accelerated recovery from radiation-induced pancytopenia in mice following exposure to radiation [33].

### 3.4. PLX-R18 Mitigates Radiation-Induced Bone Marrow Damage

Bone marrow damage and its recovery were studied through the colony-forming potential of femoral bone marrow cells and by sternal histopathological analyses and megakaryocyte quantification [33]. In the colony-forming assay (Figure 3), no colonies were detected in samples collected on Day 0 (4 h post-TBI) in either irradiated group, but colony-forming potential recovered slowly after that, with the PLX-R18-treated group showing more colonies than the PlasmaLyte-treated group on Day 14 (*p* = 0.017) and on Day 30 (*p* = 0.00016). Despite the clear improvement in colony-forming potential in the irradiated PLX-R18-treated group until Day 45, the group still showed almost 5-fold lower colony-forming potential than the non-irradiated controls on Day 45. In summary, PLX-R18 pretreatment resulted in significant recovery of radiation-induced damage in bone marrow progenitor cells.

In contrast, no irradiated PlasmaLyte-treated animals survived to Day 45. Megakaryocyte counts (Figure 4) decreased after irradiation, as expected [31,33], with a dramatic drop between Day 2 and Day 3. Megakaryocyte counts on Day 14 were not significantly different between the PLX-R18- and PlasmaLyte-treated groups, or from counts on Day 3. By Day 30, counts in the PlasmaLyte-treated group continue to decrease while counts in the PLX-R18 group were significantly higher (*p* = 0.0011) than those in the PlasmaLyte group. This data indicated that there wass a protective effect of the PLX-R18 cells to restore overall cellularity and megakaryocytes from radiation-induced injury in sternal bone marrow.

### 3.5. PLX-R18 Ameliorates Radiation-Induced Inflammation and Vascular Endothelial Damage

Biomarkers for radiation injury-induced inflammation, hematopoietic damage, and vascular endothelial damage [33,36] were quantified by ELISA from serum collected during the 30-day study (Figure 5). Hematopoietic cytokine erythropoietin (EPO) levels were high as a result of the radiation insult, but by day 14 in PLX-R18 its levels dropped significantly (*p* < 0.0001) compared to the PlasmaLyte treated group and a rapid recovery to baseline levels (as seen in the non-irradiated groups) was observed by day 30 post-TBI in PLX-R18 group. In the PlasmaLyte group though the trend was towards recovery it was significantly slower than PLX-R18 group. In the case of FMS-like tyrosine kinase 3 ligand (Flt-3L), yet another inflammation marker, recovery was observed in the PLX-R18 treated group compared to PlasmaLyte group.

In PLX-R18 group, sepsis marker Serum Amyloid A (SAA) [37] remained low throughout the study, suggesting that inflammation was prevented with the treatment of the drug. Similarly, the inflammation marker procalcitonin (PCT) levels [38] stayed low in the PLX-R18 group throughout the observation period (30 days post-TBI) whereas in the PlasmaLyte group the levels were significantly (*p* < 0.0001) higher on days 21 and 30 post-TBI.

We also analyzed for levels of matrix metallopeptidase 9 (MMP-9), a zinc metalloprotease involved in the breakdown of extracellular matrix and a known biomarker for endothelial damage [36]. Though levels decreased after radiation, no significant recovery was observed in PLX-R18 group compared to the PlasmaLyte group. Levels of the markers for vascular endothelial damage E-selectin and sP-Selectin dropped in both irradiated groups on radiation exposure. In the PLX-R18 group the recovery of these markers was very significantly rapid (*p* < 0.0001) by day 14 compared to PlasmaLyte group. In summary, we have shown that biomarkers of radiation-induced bone marrow aplasia, inflammation and vascular endothelial damages are modulated by PLX-R18 pre-treatment leading to animal survival.

### 3.6. PLX-R18 Treatment Modulated Levels of Various Inflammatory Chemokines, and Growth Factors

Serum collected from all four groups (2 non-radiated and 2 irradiated groups treated with either PLX-R18 or PlasmaLyte) were assayed for differential levels of 23 cytokines using the BioRad Luminex assay [34]. Even though most cytokine levels were elevated due to radiation injury, PLX-R18 treatment modulated the levels such that they were more towards the non-irradiated control (PlasmaLyte) (Figure 6). Granulocyte colony stimulating factor (G-CSF) levels were heightened on days 9, 14 and 21 post-TBI. PLX-R18 perturbed the levels significantly (*p* < 0.0001) compared to PlasmaLyte group. Monocyte Chemoattractant Protein-1 (MCP-1) and Eotaxin levels were elevated early, by day 3 post-TBI the difference in the two irradiated groups was significant (*p* < 0.01) where PLX-R18 treated group had lower levels.

Interleukin 1a, 2 and 6 showed similar pattern where their levels were higher in both irradiated groups as well non-radiated PLX-R18 treated group on the day of the irradiation (serum was collected 2 h post-TBI). By day 9 the levels dropped to normal levels but by day 14 the levels in irradiated PlasmaLyte treated group increased significantly (*p* < 0.0001–0.01) whereas PLX-R18 group had maintained lower levels. Both forms of Macrophage Inflammatory Proteins (MIP), MIP-1a and MIP-1b levels were high initially in all groups when compared to non-radiated control (PlasmaLyte treated). By day three the levels dropped, by day 14 only in irradiated PlasmaLyte group the levels stayed significantly high (*p* < 0.0001–0.01) whereas PLX-R18 treatment perturbed the levels to normal. The chemokine keratinocytes-derived chemokine (KC) showed slightly different patterns where there was an increase in the KC levels in irradiated PLX-R18 levels on day 3 which came down to normal. In the later time-points (day 14) the irradiated PlasmaLyte levels went up significantly (*p* < 0.0001) where PLX-R18 modulated to a lower level (Figure 6). The remaining 14 cytokines (IL-1β, IL-3, IL-4, IL-5, IL-9, IL-10, IL-12 (p40), IL-12 (p70), IL-13, IL-17A, GM-CSF, IFN-γ, RANTES, TNF-α) did not show much modulation/perturbation by PLX-R18. This data suggested that several inflammatory cytokine responses were ameliorated by PLX-R18 treatment, thus reducing systemic radiation injury.

### 3.7. PLX-R18 Abrogated Expression of Genes Involved in Autoimmune Pathways

Being an important hematopoietic organ, the spleen is a tissue of special interest. Total RNA was extracted from spleens collected from the four animal groups on Days 0 (4 h post-TBI) and 14 post-irradiation, and levels of 42 genes from autoimmune pathways were assessed by qRT-PCR (Appendix A). Fifteen genes showing differential expression upon radiation exposure and/or significant modulation by PLX-R18 are presented in Figure 7.

Though there were no significant changes observed on the day of radiation (4 h post-TBI), on Day 14, several pro-inflammatory chemokines and cytokines showed upregulation which seemed to be diminished by PLX-R18 treatment, while others showed downregulation that was ameliorated by PLX-R18. The mRNAs of two CXC chemokine receptors 1 and 2 (*Cxcr1* and *Cxcr2*) showed significant downregulation (*p* < 0.05) in irradiated animals by Day 14 compared to naïve animals, but this effect was reduced by 2 fold in the PLX-R18 group than the PlasmaLyte group. Conversely, *C3AR1* and *C4b*, which are involved in the complement system, showed higher expression in irradiated groups on Day 14, but this effect was weakened by about 2 fold in the PLX-R18-treated group. *CEBP-β* and *CD14* play important roles in macrophage function as a part of innate immunity. *CEBP-β* expression was elevated in the irradiated PlasmaLyte group by 2 fold compared to non-irradiated control on Day 14, but not in the PLX-R18 group. *CD14*, like *NOS2*, showed higher expression in irradiated groups on Day 14, but this effect was reduced by 2-fold in *CD14* and 4-fold in *NOS2* in the PLX-R18-treated group, respectively. Four *Ccl* genes (*Ccl8*, *Ccl11*, *Ccl12*, and *Ccl24*, a family of small cytokines or signaling proteins) and four interleukin cytokines (*IL-5, IL-6, IL-18, IL-23a*) showed increased expression in irradiated groups on Day 14, and this effect was reduced by 2–5 fold by PLX-R18 treatment. *Ccl11*, *Ccl12*, *IL-5*, and *IL-6* showed elevated levels in both irradiated groups on Day 0; PLX-R18 weakened this effect on *Ccl12* expression by more than 2 fold, but stayed high in *Il-5* and *Il-6*. In summary, we have shown that various pro-inflammatory cytokines/chemokines and their receptors were modulated in mouse spleen by PLX-R18 treatment before radiation exposure.

### 3.8. PLX-R18 Modulated Phosphorylation of AKT

It is known that AKT phosphorylation mediates anti-apoptotic and pro-survival events in cells [39]. Role of AKT and ERK pathways in controlling sensitivity to ionizing radiation has also been reported previously [40]. Thus, in this manuscript, expression levels of proteins from the AKT-JNK pathway in the spleen were studied by Western blot analysis (Figure 8) for days 0 and 14 post-TBI. JNK, p-38, and their phosphorylated forms showed no significant change between the non-irradiated PlasmaLyte group and either of the irradiated groups (PlasmaLyte- or PLX-R18-treated) on days 0 (4 h post-TBI) and 14 post-TBI. pAKT levels were significantly elevated on Day 0 in the irradiated PlasmaLyte group (3.5-fold higher than in the non-irradiated PlasmaLyte group), but less elevated in the PLX-R18-treated group (only 2.3-fold higher than in the non-irradiated PlasmaLyte group, again showing a significant difference (*p* = 0.02) between PLX-R18- and PlasmaLyte treatment. As pAKT levels were affected by irradiation in both treatment groups on Day 0 but not Day 14, it appeared to be an early response marker. This data indicated that AKT pathway might be involved in reducing radiation-induced damage in spleen.

## 4. Discussion

There is an ever-increasing risk to military personnel and first responders of being exposed to potentially lethal doses of ionizing radiation due to nuclear proliferation and terrorist activity that may result in entering contaminated areas for rescue operations [1]. Therefore, there is an unmet need to develop prophylactic countermeasures that can be administered to our soldiers before sending them to harm’s way. Cell-based therapeutics are available for numerous injuries and diseases [41], however there is currently no FDA approved cellular therapy as radiation countermeasure for treatment of hematopoietic acute radiation syndrome (H-ARS) further compounded by the fact that there are no radiation countermeasures approved for prophylactic use. Previously, authors have reported that cryopreserved allogeneic mouse myeloid progenitor cells (mMPC) significantly improved survival in two strains of mice irradiated with lethal doses of γ radiation (CD2F1, 9.2 Gy) and X-ray exposures (Balb/c, 9 Gy) that are known to cause acute radiation syndrome and ultimately lethality [42]. Previously we reported on the efficacy of a few promising countermeasures from various classes (γ-tocotrienol, Thrombopoietin mimetic, and the IL-11 analog, BBT-059) that protected mice from H-ARS and increased survival [31,33,43]. In the current study, we report for the first time that PLX-R18, a placenta-derived stromal cell product, is effective as a prophylactic countermeasure for radiation when administered 24 h prior to exposure to total body irradiation (TBI). We have shown that mice receiving PLX-R18 24 h before and 3 days after TBI demonstrated faster recovery from radiation-induced peripheral blood cytopenia, accelerated restoration of the clonogenic potential of femoral bone marrow, and improved megakaryocyte counts in sternal bone marrow compared to vehicle-treated animals. In addition, PLX-R18 administration attenuated biomarkers of bone marrow aplasia, sepsis, and systemic inflammation and attenuated radiation-induced inflammatory cytokines/chemokines and growth factors, including G-CSF, MIP-1a, MIP-1b, IL-2, IL-6 and MCP-1, and modulate AKT protein expression.

Administration of PLX-R18 as a prophylactic countermeasure at 2 million cells/dose IM 1 day prior to and 3 days after total body exposure to a lethal dose of γ radiation TBI led to significantly improved survival compared to vehicle-only treatment (81% vs. 23% survival; log-rank test *p* < 0.0001). This treatment regimen was also confirmed to be safe, as it did not cause significant changes to peripheral blood cell counts, body weight, or a variety of renal and hepatic biomarkers when administered to non-irradiated animals.

The protection afforded by PLX-R18 to the hematopoietic system after irradiation was evidenced by the accelerated recovery of white blood cell, lymphocyte, neutrophil, and platelet counts (most notably on Day 21 after irradiation), compared to vehicle-treated animals as observed previously by us with other promising countermeasures [32,33,43]. Although PLX-R18 treatment was unable to restore sternal megakaryocyte counts and clonogenic potential of femoral bone marrow cells to the levels observed in non-irradiated animals, both parameters were significantly higher by Day 30 after irradiation in PLX-R18-treated animals than in PlasmaLyte-treated counterparts, demonstrating the ability of PLX-R18 to aid in the recovery of femoral and sternal bone marrow stem cells. Furthermore, femoral bone marrow clonogenic potential assessed on Day 45 in the PLX-R18-treated group demonstrated continued improvement of this parameter.

The physiological response to radiation injury may cause a wide range of systemic responses due to damage incurred in multiple tissues and organs. Following radiation, macrophages produce a variety of cytokines and growth factors, which in turn, initiate a wide range of systemic responses via the central nervous system by increasing production and differentiation of bone marrow cells and expression of acute phase proteins [37]. Ossetrova et al. demonstrated dose-and time dependent changes in hematopoietic cytokines and acute phase proteins Flt3L, IL-6, G-CSF, TPO, EPO, and SAA in mice serum exposed to total body radiation which may function as prognostic biomarkers of ARS [37]. EPO has been shown to play major role in regulating erythropoiesis and promotes survival, proliferation, and differentiation of erythroid progenitor cells. Flt3L has been reported as an efficacy biomarker of bone marrow damage and SAA and PCT was reported as marker of sepsis following organ damage [37,38]. All these protein levels are significantly increased after radiation with time as expected, however, PLX-R18 pre-treatment lowered these protein levels and brought them back to near basal levels. This suggests that part of the radioprotection afforded by PLX-R18 is the result of reducing bone marrow damage and sepsis following radiation exposure in mice [33,38].

Here, we have shown that various cytokines were upregulated in irradiated mice that received the vehicle alone. Conversely animals that received PLX-R18 had cytokine levels that stayed at our near non-irradiated levels as observed previously [34]. Cytokines such as MCP-1 and Eotaxin which are responsible for stimulation of monocyte and macrophage production [26], were elevated shortly after exposure. Other cytokine such G-CSF, IL-1, IL-2, IL-6, MIP-1α, and MIP-1β, were upregulated at later time points (~day 9 onward). G-CSF functions to promote bone marrow production, members of the Interleukin family are responsible for inflammation response, and members of the MIP family are secreted by macrophages as an inflammation response [44]. Taken together, these represent an early response of increased macrophage production followed by an activation of inflammation response pathways in the animals administered the vehicle, likely in compensation of the injuries resulting from exposure. In the animals administered PLX-R18 this response was not observed indicating that the injury was less severe. These results indicate that these secreted cytokines may play a role in survival from hematopoietic acute radiation damage in animals pre-treated with PLX-R18.

Radiation-induced vascular damage plays a critical role in the mechanism of early radiation responses in many different organ systems [36]. Peripheral blood levels of the endothelial cell E-selectin, P-selectin, and the matrix metalloproteinase 9 (MMP-9) are commonly used to monitor changes in endothelial function [36]. Favorable effect on endothelial markers E-selectin and P-selectin were observed in mice treated with PLX-R18 indicating protective properties of PLX-R18 extending to vasculature in addition to protecting the hematopoietic system as seen in levels of various cytokines [35]. On the other hand, levels of the MMP-9 which is a regulatory factor in neutrophil migration across the basement membrane did not show any change with PLX-R18 treatment [36]. Therefore, it is likely that some of the radioprotective properties of PLX-R18 may occur due to recovery from endothelial dysfunction of normal tissues.

It is known that AKT phosphorylation mediates anti-apoptotic and pro-survival events in cells [39]. We investigated the AKT-JNK pathway proteins levels in mouse spleen following exposure to total body radiation which causes damage in hematopoietic organs and tissues. We found levels of phosphorylated AKT being abrogated by PLX-R18 pre-treatment at an early time point, 4h post radiation exposure. The role of AKT and ERK pathways in controlling sensitivity to ionizing radiation has been reported previously [40]. Our data suggests that the AKT pathway might have a role shortly after radiation exposure to protect normal tissue damage in the presence of PLX-R18. 

## 5. Conclusions

In conclusion, our data show the systemic effect of PLX-R18 pre-treatment protects mice from radiation-induced bone marrow aplasia and rescues them from lethality. In addition to attenuating hematopoietic and immune system dysfunction, PLX-R18 also reduces vascular oxidative stress and pro-inflammatory cytokines/chemokines and growth factors, inhibited the expression of pAKT. Further studies need to be conducted to explore the detailed mechanism and cascade of events contributing to the radioprotection afforded by PLX-R18 cells. Evaluating the efficacy of this countermeasure on female mice would also add value to the studies that could be included for an FDA IND application. Further development of PLX-R18 as a radioprotectant could have a significant impact on radiation biology, radiation oncology, and military medicine.

## Figures and Tables

**Figure 1 genes-13-01756-f001:**
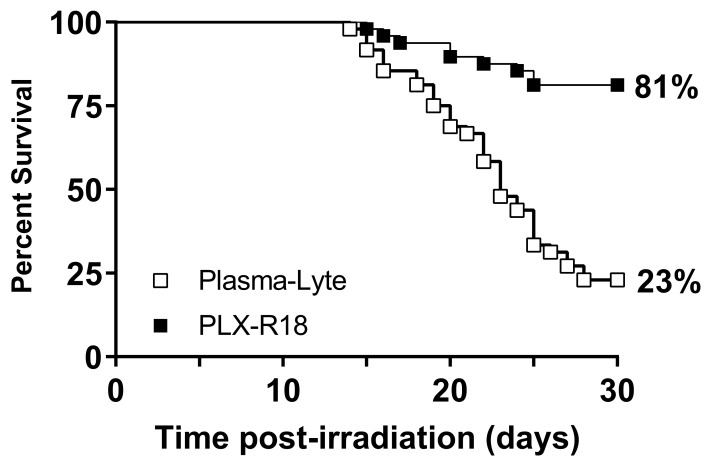
Survival of C57BL/6 male mice following total-body irradiation with 8 Gy at an estimated rate of 0.6 Gy/min and IM administration of two doses of PLX-R18 (2 million cells/dose) on 1 day prior to and 3 days post-TBI (■) or PlasmaLyte as vehicle (□).

**Figure 2 genes-13-01756-f002:**
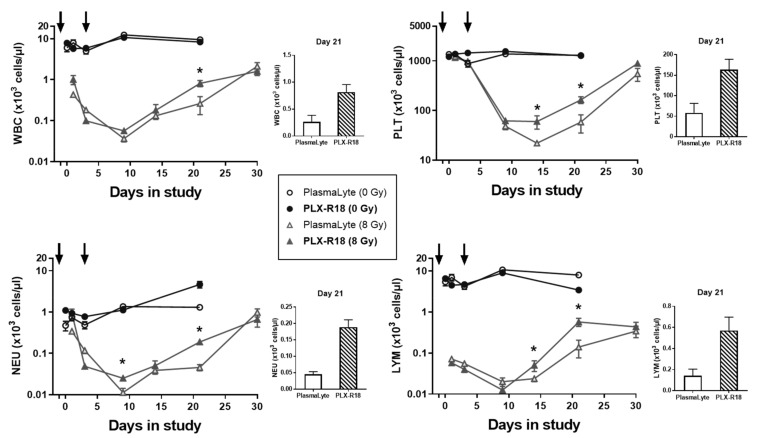
Recovery of peripheral blood cells [white blood cells (WBC), neutrophils (NEU), platelets (PLT), and lymphocytes (LYM)] of unirradiated mice treated with vehicle PlasmaLyte (○) and PLX-R18 (●) and irradiated (8 Gy) mice treated with PlasmaLyte (∆) and PLX-R18 (▲). Either PlasmaLyte or PLX-R18 (2 million cells/dose) was administered (IM) 24 h prior to and 3 days post-TBI (indicated by arrows). Day 0 represents day of irradiation. Data represented are mean ± standard error of the mean (SEM) for n = 8 mice. Significant difference (*p* < 0.005–0.0125) between PLX-R18-treated and vehicle-treated irradiated group by ANOVA is indicated with an asterisk (*). Some data points in the figure do not have error bars that are visible because they are smaller than symbols. Bar graphs show the most significant data (day 21 post-TBI) for each of the cell type.

**Figure 3 genes-13-01756-f003:**
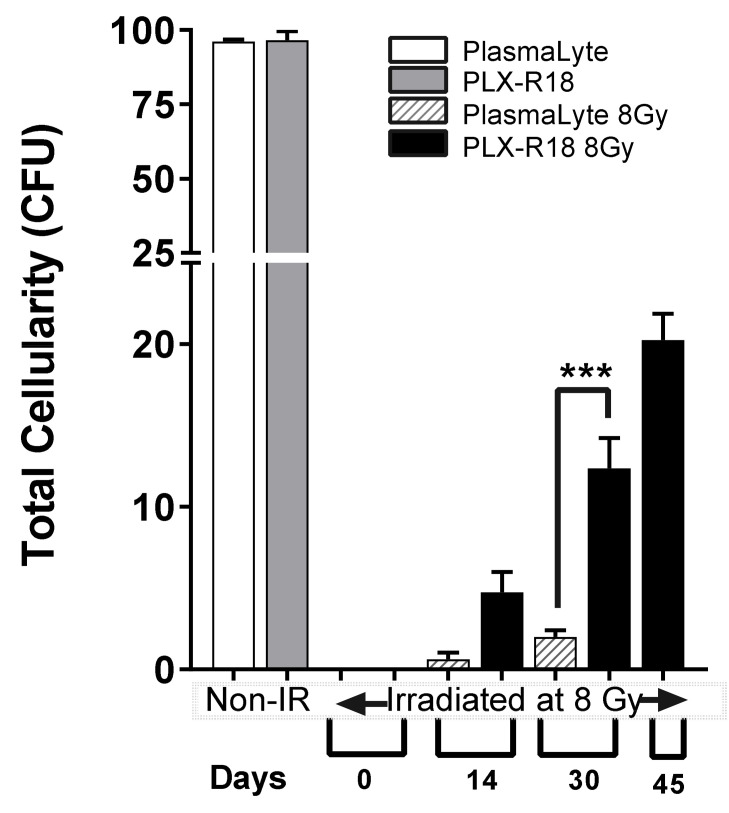
Effect of 8 Gy TBI on femoral bone marrow. Clonogenic potential of bone marrow cells was assessed by a CFU assay. Colony forming units (CFU) were assayed on days 1, 14, 30 and 45 after exposure. Cells from three femurs were pooled, counted, and each sample plated in duplicate to be scored 14 days after plating. Data are expressed as mean ± Standard error of mean (SEM). No statistically significant difference was determined between irradiated and naïve groups. Non-irradiated but administered with either PlasmaLyte or PLX-R18 were used as controls. *** indicates *p* = 0.00016.

**Figure 4 genes-13-01756-f004:**
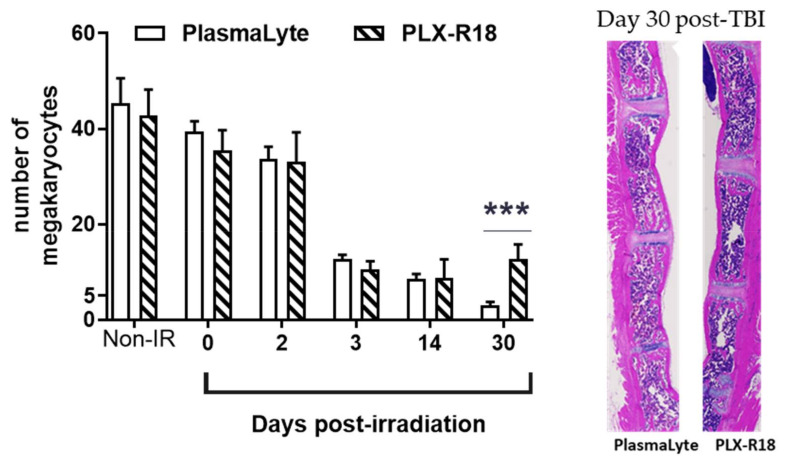
PLX-R18 treatment promoted sternal bone marrow hematopoietic cell recovery after lethal exposure of 8 Gy TBI when administered first dose at 24 h prior to and 3 days post-TBI. Bone marrow megakaryocyte numbers were quantitated from histological sections from days 0 (2 h post-TBI), 2, 3, 14 and 30 post-TBI. Significant increase (indicated as *** *p* = 0.0011) in bone marrow cellularity and megakaryocytes were observed on day 30 post-TBI in the PLX-R18 treatment group. Data represented are mean ± standard error of the mean (SEM) for n = 8 mice. Representative H&E stained sternal bone marrow sections are shown for PlasmaLyte and PLX-R18 on day 30 post-TBI.

**Figure 5 genes-13-01756-f005:**
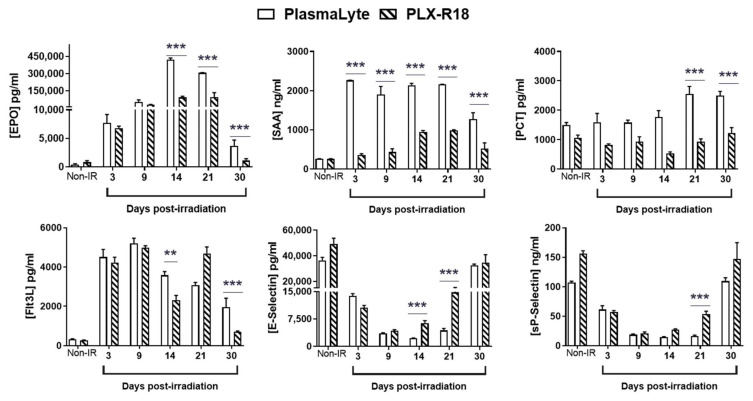
PLX-R18 restores levels of endothelial and radiation injury biomarkers. PLX-R18 administration inhibited radiation-induced elevated synthesis of various biomarkers in mouse serum when compared to vehicle-treated animals. The levels of EPO, SAA, PCT, Flt3L, E-selectin and sP-selectin were evaluated from serum from samples collected on days 3, 9, 14, 21 and 30 post-TBI by ELISA. Data represented are mean ± standard error of the mean (SEM) for n = 8 mice per group; *** *p* < 0.0001; ** *p* = 0.002.

**Figure 6 genes-13-01756-f006:**
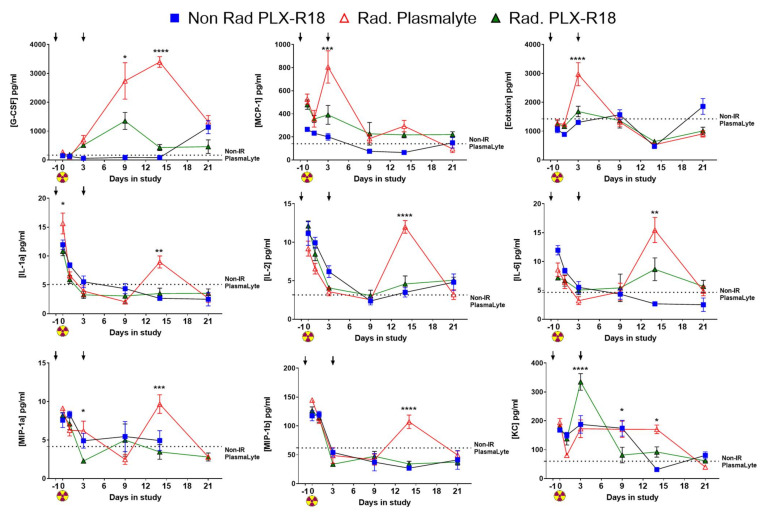
Differential levels of circulatory cytokines & recovery by PLX-R18 by Luminex assay. Serum samples were subjected to an array of 23 cytokines. The four murine proteins were significantly elevated (n = 5/time point). * *p* < 0.05, ** *p* < 0.01, ***/**** *p* < 0.0001. That yellow symbol means the day of radiation. The arrow means the days we administered the PLX-R18.

**Figure 7 genes-13-01756-f007:**
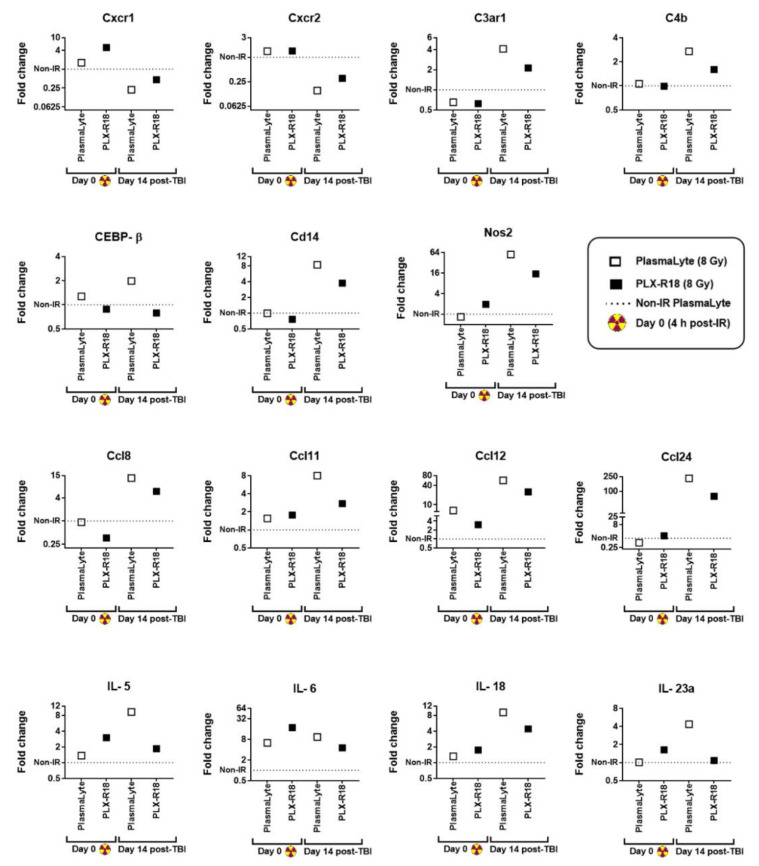
Differential levels of cytokines & receptors from the autoimmune pathway as early and late markers of radiation injury and modulation by PLX-R18 treatment by PCR. Spleen samples were subjected to an array of 42 genes.

**Figure 8 genes-13-01756-f008:**
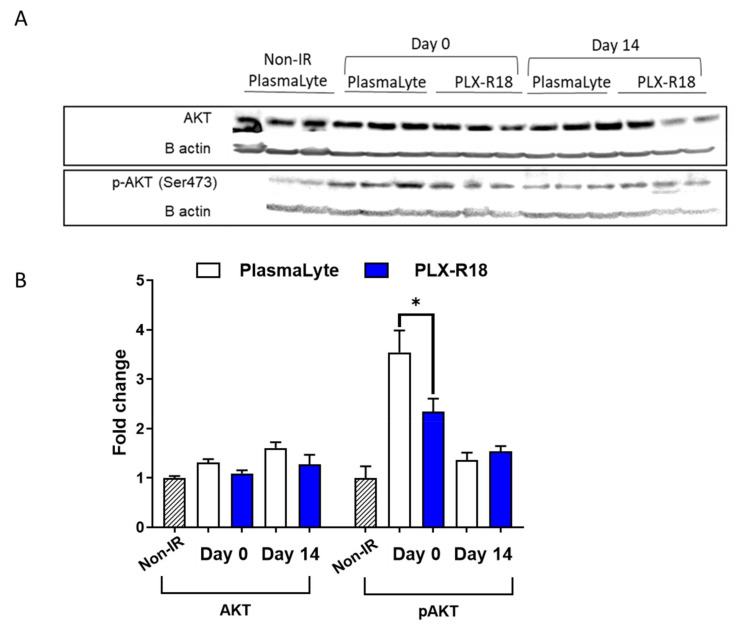
Differential levels of proteins and their phosphorylated forms as markers of inflammation due to radiation injury and effective amelioration by PLX-R18 treatment. Levels of AKT and pAKT were evaluated by Western blot analyses (**A**). The band density was quantified using image analyses software from BioRad (**B**) * indicates *p* < 0.05.

## Data Availability

Not applicable.

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
