# Peer review of "Pre-Administration of PLX-R18 Cells Protects Mice from Radiation-Induced Hematopoietic Failure and Lethality"

_genes, 2022, doi:10.3390/genes13101756_

Round 1
Reviewer 1 Report
This a very well-written manuscript with sound experimental design, logical conclusions based on the robust data. In this study, the authors investigated the role for PLX-R18 as a prophylactic agent to protect from radiation-induced bone marrow failure and lethality in C57BL/6 male mice.
They showed that PLX-R18 promotes increased post-radiation survival, increased recovery of WBCs (neutrophils, lymphocytes), platelets and RBCs, attenuated biomarkers of inflammation, sepsis and endothelial dysfunction and Akt signaling pathway. PLX-R18 treated mice showed increased clonogenic potential of bone marrow cells.
Some minor points that need to be addressed are:
1. In Fig. 4, the H&E staining of sternum sections is missing. Please provide details on quantification of megakaryocytes, at what magnification were specimens analyzed, and how many fields per section/mouse etc. were analyzed.
Author Response
In Fig. 4, the H&E staining of sternum sections is missing. Please provide details on quantification of megakaryocytes, at what magnification were specimens analyzed, and how many fields per section/mouse etc. were analyzed.
Response: Thanks for the review. We have revised the Figure 4 with the representative image of the H&E stained section.
Megakaryocytes quantification was carried out by averaging the number of cells per 10 (40X) high-powered fields (HPFs). We have added this information to the methods section 2.6.3.
Reviewer 2 Report
This paper studied the protective role of PLK-R18 cells against radiation-induced hematopoietic cell lethality. PLK-R18 administration facilitated rapid recovery of blood cell number by attenuating autocrine/paracrine factors. Authors also indicated no side-effect of PLK-R18 administration in mouse model. Overall, the paper is well written.
Major issue
1. Indicate approval number for animal experiments from Institute Animal Ethics Committee.
2. In figure 6, inflammation response is peak at day 14 after TBI, while number of lymphocytes still decreased. Which cells are expected to generate cytokins?
Author Response
- Indicate approval number for animal experiments from Institute Animal Ethics Committee.
Response: Thanks for bringing this up. There is a possibility that the manuscripts were sent for review prior to our submission of the revised the manuscript with addition of the Institutional Review Board Statement. We have that section towards the end of the manuscript in the revised version.
- In figure 6, inflammation response is peak at day 14 after TBI, while number of lymphocytes still decreased. Which cells are expected to generate cytokins?
Response: We are measuring circulatory levels of various cytokines in peripheral blood. The differential levels of cytokines after irradiation could be a result of a multilineage hematopoietic progenitor cells in the bone marrow and other hematopoietic tissues at this radiation dose. This in turn possibly resulted in the hematopoietic rescue and survival post-irradiation in animals pre-treated with PLX-R18 cells.
Reviewer 3 Report
It was a pleasure to review the manuscript "Pre-administration of PLX-R18 cells protects mice from radiation-induced hematopoietic failure and lethality". This manuscript raises some interesting points. However, there are several major and minor concerns regarding the experimental design and subsequent analyses.
major concerns:
1) In the Results section, the experimental results elaborate too much. The results should be explained briefly. And there should be a small summary of the results in each result section.
2) In the Fig3, 5 and 6, there is too much repetition between the figure legend and the text. Don't repeat the results of the experiment in the figure legend.
3) What are these 14 cytokines in Line 410 on Page 12? Why did the remaining 14 cytokines not show much modulation/ perturbation by PLX-R18? Why did you test so many cytokines? Should be explained clearly.
4) In the Result 3.8, why did you determine the protein expression of AKT? What does AKT have to do with PLX-R18 cells of protecing mice from radiation-induced hematopoietic failure and lethality? Should be explained clearly.
5) In the conclusion section, the authors repeated the experimental results too much and did not have a real discussion. The section need major revision.
6) In the whole manuscript, the authors only found the radiation protection phenomenon of PLX-R18 cells, but did not study its mechanism. The related mechanisms involved in PLX-R18 cells protects mice from radiation-induced hematopoietic failure and lethality should be explored.
minor concerns:
1) In the FigS1, there is no A, B and C annotation. The symbol marks of each experimental group in the FigS1 are not consistent with those described in the figure legend. And the description of B and C in the text is not consistent with those described in the figure legend.
2) There are mistakes in the full names of AST and ALT in Line 265-266 on Page 6, the two names are identical.
3) In the Fig4, what does # on the vertical axis mean?
4) In the Fig8B, the symbol marks of experimental group are not simple and clear enough for people to understand.
5) I didn’t see representative H&E stained sternal bone marrow sections discribed by the authors.
Author Response
major concerns:
- In the Results section, the experimental results elaborate too much. The results should be explained briefly. And there should be a small summary of the results in each result section.
Response: Thanks for the review. The Results sections are revised.
2) In the Fig3, 5 and 6, there is too much repetition between the figure legend and the text. Don't repeat the results of the experiment in the figure legend.
Response: As per your suggestion we have revised the figure legends of Fig 3, 5 and 6 by deleting the results from these sections.
3) What are these 14 cytokines in Line 410 on Page 12?
Response: We have added the names of the 14 cytokines levels those were not modulated by PLX-R18 treatment in the Results section 3.6.
4) Why did the remaining 14 cytokines not show much modulation/ perturbation by PLX-R18? Why did you test so many cytokines? Should be explained clearly.
Response: We have used an immunoassay kit (Bio-Plex Pro Mouse Cytokine 23-Plex Immunoassay by BIORAD) comprising of 23 cytokines involved in the inflammatory pathways. We used this 23-plex to find which cytokines are modulated by PLX pre-treatment. We have added some explanation in the discussion section.
5) In the Result 3.8, why did you determine the protein expression of AKT? What does AKT have to do with PLX-R18 cells of protecing mice from radiation-induced hematopoietic failure and lethality? Should be explained clearly.
Response: It is known that AKT phosphorylation mediates anti-apoptotic and pro-survival events in cells (Tessner et al, The journal of Clinical Investigation, 2004, doi:10.1172/JCI22218). We investigated AKT-JNK pathway proteins levels as mentioned in Results section 3.8. We found levels of phosphorylated AKT being abrogated by PLX-R18 pre-treatment at early time point, 4h post radiation exposure. Role of AKT and ERK pathways in controlling sensitivity to ionizing radiation has also been reported previously (Sun Parka et al, 2015, doi.org/10.1016/j.ejcb.2015.08.003).
6) In the conclusion section, the authors repeated the experimental results too much and did not have a real discussion. The section need major revision.
Response: Thank you for the insightful comment. We have revised the discussion section.
7) In the whole manuscript, the authors only found the radiation protection phenomenon of PLX-R18 cells, but did not study its mechanism. The related mechanisms involved in PLX-R18 cells protects mice from radiation-induced hematopoietic failure and lethality should be explored.
Response: We thank the reviewer for taking time to review the manuscript. We understand that the related mechanisms involved in PLX-R18 cells protection of mice from radiation-induced hematopoietic failure and lethality should be explored. Detailed mechanistic studies are part of our on-going studies and the data will be published on completion.
minor concerns:
- In the FigS1, there is no A, B and C annotation. The symbol marks of each experimental group in the FigS1 are not consistent with those described in the figure legend. And the description of B and C in the text is not consistent with those described in the figure legend.
Response: Thanks for noticing that. We have revised the figure and the figure legend.
Round 2
Reviewer 3 Report
I am very pleased the revised manuscript. Authors made careful revision according the reviewers’s opinions. However, I have some suggestions and points that the authors should consider.
1) I pointed out that “In the Results section, the experimental results elaborate too much. The results should be explained briefly. And there should be a small summary of the results in each result section.” But, the Results was unchanged in the revised manuscript.
2) In the Result 3.8, the reason that you determined the protein expression of AKT should be explained in the revised manuscript. Thus, it helps readers to understand.the study.
3) In the Fig4, from a rigorous point of view, I think that the meaning of # on the vertical axis should be expressed in words but not the symbol.
Author Response
I am very pleased the revised manuscript. Authors made careful revision according the reviewers’s opinions. However, I have some suggestions and points that the authors should consider.
Response: We would like to thank you for your time and efforts in reviewing the manuscript.
- I pointed out that “In the Results section, the experimental results elaborate too much. The results should be explained briefly. And there should be a small summary of the results in each result section.” But, the Results was unchanged in the revised manuscript.
Response: Thanks for the suggestion. We have extensively revised the results sections and also added “In summary” statement at the end of each result section.
2) In the Result 3.8, the reason that you determined the protein expression of AKT should be explained in the revised manuscript. Thus, it helps readers to understand.the study.
Response: We have added that in the beginning of the result 3.8.
- In the Fig4, from a rigorous point of view, I think that the meaning of # on the vertical axis should be expressed in words but not the symbol.
Response: We have revised the figure. In the past we have published many manuscripts using this symbol in graphs including our most recent publication DOI: 10.1038/s41598-022-07426-7.